# Properties and Reaction Mechanisms of Magnesium Phosphate Cement Mixed with Ferroaluminate Cement

**DOI:** 10.3390/ma12162561

**Published:** 2019-08-11

**Authors:** Liang Jia, Fangli Zhao, Jian Guo, Kai Yao

**Affiliations:** 1College of Civil Engineering, Lanzhou University of Technology, Lanzhou 730050, China; 2Department of Civil & Environmental Engineering, National University of Singapore, Singapore 117576

**Keywords:** magnesium phosphate cement, ferroaluminate cement, properties, reaction mechanisms

## Abstract

A certain amount of ferroaluminate cement (FAC) was substituted for MgO during the magnesium phosphate cement (MPC) preparation to obtain the MPC–FAC composite cement. The influence of FAC on the strength, water resistance, pH, and setting time of MPC–FAC composite cement were examined. The microstructure and chemical composition were also analyzed by adopting scanning electron microscopic energy-dispersive spectrometer and X-ray diffraction, respectively. The study showed that setting time of MPC–FAC composite cement was dramatically prolonged when FAC substitution for MgO was between 30 and 40 wt %. The strength of MPC–FAC did not decrease during the early curing time (1 h and 1 d), whereas it increased during the late curing time (3, 7, and 28 days). Moreover, the existence of FAC decreased the hydrated product K-struvite during the early curing time and thus dramatically enhanced the water-resistance of MPC–FAC. With the addition of FAC, a large number of cementitious materials of AFt and AFm, as well as flocculent colloidal substances of AH_3_, C–S–H, and FH_3_, were generated during the hydration of MPC, which were filled in the internal pore of the hydrate. Thus, the internal compactness of the sample increased, while the compact protective covering layer was generated on the surface to enhance the water resistance and strength in the late curing time.

## 1. Introduction

Magnesium phosphate cement (MPC) is a water-hardening inorganic cementitious material [1,2,3]. Compared with the conventional Portland cement, MPC has different chemical reaction, product, and performance. MPC has many advantageous properties such as high early-strength, small drying shrinkage, excellent adhesive performance, and good corrosion resistance and frost resistance [4,5,6,7,8,9,10], and thereby is used for infrastructural projects worldwide, such as airfield runways, bridges, highways, and patching damaged pavements [11,12,13]. Also, it is often applied in the stabilization of mixed wastes [14,15,16] as a bone-repair material [17,18,19]. Nonetheless, it has some drawbacks as a rapid-repair material, such as too short setting time and poor water resistance [13,20].

In order to improve the properties of MPC, the researchers conducted a series of studies. The main method was to add mineral admixture as a modifier to the MPC, such as fly ash [21,22,23,24], metakaolin [7,25], steel slag [26], and fiber [27,28]. Gardner et al. placed the aluminosilicates into MPC paste and reported its dissolution characteristics [29]. Yang et al. explored the change in the performance of MPC mortar with different steel slag powder content [26]. The existence of a small amount of metakaolin in MPC mixture leads to a favorable influence on its setting behavior, which provides a significant enhancement in the water-resistance and mechanical properties [30,31,32,33,34]. An improvement in active effect of slag powder, filling materials, and particle size distribution results in an obvious growth in the early compressive strength [26]. Jiang et al. tested the water resistance and mechanical properties of MPC by preparing MPC mortar samples with various steel slag powder contents [26]. The results showed that the incorporation of steel slag powder causes a decrease in the early strength but an increase in water resistance. Gai et al. investigated the effect of cellulose content on water stability and mechanical strength, and the results indicated that an increase in cellulose content leads to an enhancement in the water stability while a decrease in the mechanical strength of MPC mortar [35]. The aforementioned studies showed that the improvement effect of mineral admixture on MPC performance depended on its type and physical and chemical properties.

In addition, studies have shown that the water-hardening material can enhance the water resistance of MPC. Different types of cement were added to MPC to form a composite system. Calcium aluminate cement (CAC) is a new type of cement. To determine the effect of the addition of CAC on water-resistance and strength of MPC, Zhang et al. conducted a series of studies on CAC–MPC mixture. They found that the introduction of CAC creates a certain amount of hydrous and amorphous phases, which coated the K-struvite crystals, and resulted in better water-resistance and a higher compressive strength of the mixture [36]. Li et al. explored the influence of sulfoaluminate cement (SAC) on the performance of MPC [31,37]. The results showed that the existence of SAC provides a significant enhancement on the water stability and compressive strength of MPC. However, sulfoaluminate cement had a limited effect on the growth of water resistance. The reason is mainly related to the amount of ettringite in the MPC–SAC mixture, in which a lower PH value (less than 11) inhibits the formation of ettringite, and thus did not present an obvious growth on the water resistance [36]. Based on the influence of SAC on MPC, it was believed that a similar effect or better improvement can be achieved when the high alkaline aluminoferrite cement is mixed with MPC. Ferroaluminate cement (FAC) is a series of SAC. The main minerals of FAC are 4CaO·3Al_2_O_3_·SO_3_ (C_4_A_3_S), Ca_2_SiO_4_(C_2_S), and C_6_AF_2_ [38]. The products formed by FAC hydration mainly include Ca_6_Al(SO_4_)_3_(OH)_12_·26H_2_O (AFt), AH_3_ gel, Ca(OH)_2_, C–S–H gel and FH_3_ gel. The hydration process of the aluminoferrite cement produced a large amount of Ca(OH)_2_, which was partially consumed in the formation of secondary hydrates, and a small amount is present in the final formation of hydrated bodies. The existence of Ca(OH)_2_ creates a relatively high liquid phase alkalinity of the hydrated body and the pH value is usually between 12 and 13 [39,40].

In this study, a certain amount of FAC was substituted for MgO to obtain the MPC–FAC composite cement. The mechanism of the prolongation setting time and the enhanced water resistance was studied. Besides, X-ray diffraction (XRD) and scanning electronic microscopic energy-dispersive spectrometer (SEM-EDS) were adopted to study the reaction process and reaction products of the new MPC–FAC specimens.

## 2. Experimental Section

### 2.1. Materials and Mixture Proportion

The studied materials included the potassium dihydrogen phosphate (KH_2_PO_4_) powder, high-temperature calcination magnesia (MgO) powder, potassium dihydrogen phosphate (KH_2_PO_4_) powder, borax (Na_2_B_4_O_7_·10H_2_O) powder, and purified water. Magnesia powder was calcined from magnesite at 1600 °C for 6 h. The magnesia, potassium dihydrogen phosphate, and borax powder were supplied by Guangzhou Suixin chemical factory, Guangdong, China. The ferroaluminate cement was provided by Hangzhou Yuewei building materials Co., Ltd., Hangzhou, China. The chemical compositions (by mass) of the ferroaluminate cement were determined by X-ray fluorescence spectrometry and are listed in Table 1.

Mix proportions of the prepared MPC–FAC mortar are shown in Table 2. FAC/MgO ratio ranged from 0 to 1, the water/binder ratio and the sand/binder ratio were 0.22 and 1, respectively, and were selected for the sample preparation.

### 2.2. Specimen Preparation

MgO, borax, potassium dihydrogen phosphate, ferroaluminate cement, and sand were first measured by mass according to the predesignated mix ratio. After that, the dry mixture was mixed using a cement mortar mixer for 30 s. The required amount of water was then added to the dry mixture and the mixing process was continued until a uniform mixture was achieved. The mixture was then prepared into a cuboid mortar by pouring it into a mold with dimensions of 40 × 40 × 160 mm^3^ in the laboratory at a humidity of above 90% and relative temperature of 20 ± 2 °C. The molds were then wrapped with a plastic film and stored for 30 min [41]. Subsequently, the specimens were demolded and then recovered with the plastic filmand cured for 1 h, 1 day, 3 days, 7 days, 28 days for determining the compressive strength and flexural strength, and cured for 7 + 7 days, and 7 + 28 days (the specimens were first air-cured for 7 and then immersed in water for 7 day and 28 days, respectively) for testing water resistance, also, the compressive strength and flexural strength were measured.

### 2.3. Testing Methods

#### 2.3.1. Setting Time

The initial setting time was measured using a Vicat needle following the procedures described in the Chinese National Standard GB/T 1346-2011 [42]. An MPC–FAC fresh paste was used to determine the mixture setting time. The test data was collected for every 30 s, and then the setting time of the specimen was observed for every 10 s when it was close to the initial setting time [36]. Since the interval time of MPC between the initial setting time and the final setting time is very short, the initial setting time was taken as the setting time.

#### 2.3.2. pH Value

Pastes with a deionized water-to-binder ratio of 10 were intended to test the pH value of the mixture. A pH meter (Model/MIK-PH5022, Meacon, Berlin, Germany) was used to measure the pH value. During the test, the pH values of the pastes were recorded every 30 s in the first 1 h of the hydration process. Thereafter, it was recorded every 1 h until 7 days [36].

#### 2.3.3. Compressive Strength and Flexural Strength Test

The compressive strength and flexural strength of the MPC–FAC mortar cuboids were examined for various ages of 1 h, 1 d, 3 days, 7 days, 28 days, 7 + 7 days, and 7 + 28 days using the direct-drive electrohydraulic servo system in material testing machine with a loading rate of 1 mm/min in accordance to the BS EN 12390-3: 2009 [43]. The strengths of the MPC–FAC mortar samples with three replicates were tested according to predetermined ages.

#### 2.3.4. Water Resistance

The strength retention coefficient (*K*) was used to evaluate the water resistance of MPC–FAC, which was measured following the method described in the Chinese National Standard (GB/T 24312-2009) [44]. The compressive strength and flexural strength of MPC–FAC mortar were tested after 7 days of air curing [45]. After the sample preparation, the samples were wrapped with a plastic film and air-cured for 7 days. Subsequently, the plastic film was removed and the samples were cured for 7 days and 28 days in water, respectively. The value of *K* can be calculated by the Equation (1):(1)$$K=\frac{f}{F}$$
where *f* is the flexural strength and compressive strength of MPC–FAC mortar that went through 7-day air curing and was then soaked in water for 7 days and 28 days, respectively; and *F* is the strength of the specimens cured in the air with the plastic film for 7 days.

#### 2.3.5. XRD Analysis

An XRD-7000X diffractometer (Shimadzu Co., Tokyo, Japan) with Cu Ka radiation in the 2*θ* range from –12° to 164° was used to analyze the compositions of the MPC paste, FAC paste, and MPC–40FAC paste. At the predetermined stage, the specimens were ground and then washed and filtered in absolute anhydrous ethanol three times to terminate their hydration. Subsequently, they were dried in a vacuum dryer at 40 °C and finally sealing until X-ray diffraction analysis.

#### 2.3.6. SEM Analysis

Freshly fractured pieces of the MPC paste and MPC–40FAC paste were selected, dipped in absolute anhydrous ethanol for 24 h to stop the hydration, vacuum dried at 60 °C, and then gold coated [37]. The gold-coated sample was observed under a scanning electron microscope (Model JSM-5600LV, Japan electron optics laboratory, Tokyo, Japan). Also, the compositions of the area of interest were quantitatively analyzed via EDS.

## 3. Results and Discussion

### 3.1. Setting Time of MPC–FAC Paste and Compressive Strength of MPC–FAC Mortar at the Age of 1 h

The effect of the FAC content on the setting time of MPC–FAC pastes are shown in Figure 1. With an increase of FAC content, the setting time of the MPC–FAC pastes gradually increased from 15 min to 25 min. The increase of the setting time of the MPC–FAC pastes is mainly attributed to a greater setting time of the FAC paste than that of MPC, which indicated that the primary reaction in the MPC–FAC paste was the reaction between phosphate and magnesium at the early hydration stage.

The compressive strength of the MPC–FAC mortar at the curing age of 1 h is shown in Figure 2. With the increment in the substitution ratio of FAC for MgO, the compressive strength of the MPC–FAC mortar did not change a lot; whereas the compressive strength of the MPC–FAC mortar at 1 h of curing decreased suddenly when the substitution ratio of FAC for MgO exceeded 40 wt %. Thus, in the preparation of MPC–FAC, the substitution of FAC for MgO could prolong the mortar coagulation time and guarantee the early strength not to decrease with the appropriate substitution ratio, which was an effective procedure to solve the problem of slow coagulation and early strength.

### 3.2. pH Value

The pH values of the MPC paste and MPC–40FAC paste are shown in Figure 3. The rise in the pH values of the specimens was faster due to the dissolution before 1 h as shown in Figure 3a. However, the pH values of the MPC paste demonstrated a fast decrease and then increased slowly. After 1 day of hydration, the pH values of specimens increased slowly with time, as shown in Figure 3b. The pH value of MPC–40FAC was greater than that of MPC. Specifically, the pH values of MPC–40FAC and MPC were more than 12.5 and around 11.5, respectively.

### 3.3. Compressive Strength and Flexural Strength

The compressive strength and flexural strength of the MPC mortar and MPC–FAC mortar with different FAC contents are shown in Figure 4 and Figure 5 at different ages of 1 h, 1 d, 3 days, 7 days, and 28 days. For the reference MPC mortar, the compressive strength of the MPC–FAC mortar first slightly decreased and then increased with the increase in FAC content at ages of 1 h, 1 d, and 3 days. The compressive strengths of MPC–30FAC and MPC–40FAC were higher at an early stage. Especially, the compressive strength was 11.57 MPa, 29.58 MPa, and 29.58 MPa at the ages of 1 h, 1 d, and 3 days, respectively. However, when the additional dosage of FAC was more than 40 wt %, the compressive strengths of MPC–FAC at all the curing time decreased significantly.

Figure 5 shows that the developing trend of the flexural strength was almost the same as that of the compressive strength. At the curing age of 1 h, the flexural strength of the MPC–FAC mortar was lower than that of MPC mortar, wherein the flexural strengths of MPC–30FAC mortar and MPC–40FAC mortar were larger (2.86 MPa and 3.01 MPa, respectively). At the curing ages of 1 day and 3 days, the flexural strengths of the MPC–10FAC mortar and MPC–20FAC mortar were all lower than that of MPC, whereas the flexural strengths of MPC–30FAC mortar and MPC–40FAC mortar were all higher than that of MPC mortar. Besides, the flexural strength of MPC–50FAC was very small. At the ages of 7 days and 28 days, the flexural strength of the MPC–FAC mortar increased initially and then decreased with the increase in the substitution ratio of FAC for MgO. With the substitution ratio of FAC for MgO between 30 and 40 wt %, the flexural strength and the compressive strength of the MPC–FAC mortar were relatively larger at curing ages of 3 days, 7 days, and 28 days.

### 3.4. Water Resistance

The attenuation in strength in water is mainly attributed to the poor water resistance of the material. [46,47,48,49,50]. Thus, to evaluate the water resistance of MPC and MPC–FAC composite cement, the strength retention ratio *K* was introduced in this study as the evaluation index for water resistance. The compressive strength and flexural strength retention coefficient of MPC mortar and MPC–FAC mortar with different FAC contents are shown in Table 3 and Table 4 at different curing ages of 7 + 7 days and 7 + 28 days.

The flexural strength and compressive strength of MPC mortar decreased after water immersion. The compressive strength retention coefficient after water immersion for 7 days and 28 days was 0.8998 and 0.7759, respectively. Besides, the corresponding flexural strength retention coefficient was 0.9245 and 0.7962, respectively. The flexural strength and compressive strength of the MPC–FAC mortar after water immersion were higher than those before water immersion, with a strength retention coefficient larger than 1. This showed that the water resistance of the MPC–FAC cement was far better than that of the MPC cement. Table 3 shows that the compressive strength of the MPC–FAC mortar increased initially and then decreased with the increase in the substitution ratio of FAC for MgO. With the substitution ratio of FAC for MgO between 30 and 40 wt %, after 7 + 7 days and 7 + 28 days of water immersion curing of MPC–FAC, the corresponding flexure and the compressive strength were all higher. After water immersion of MPC–30FAC and MPC–40FAC mortar, the compressive strength was the highest, corresponding to the strength retention ratio of 1.2149 and 1.1342 for 7 + 7 days, respectively, and of 1.2627 and 1.2343 for 7 + 28 days, respectively. Table 4 shows that the flexural strength of the MPC–FAC mortar after water immersion increased with the increase in the substitution ratio of FAC for MgO. The main reason was the more compact structure of the MPC–FAC mortar due to the addition of FAC. Besides, FAC continued to undergo hydration reaction in water immersion curing, which increased the strength of the MPC–FAC mortar.

### 3.5. Reaction Mechanisms of MPC Mixed with Ferroaluminate Cement

The analysis of the strength and water resistance of the MPC–FAC mortar revealed that partial substitution of FAC for MgO can prolong the coagulation time, improve the mechanical performance, and solve the poor water resistance of MPC. To further study the mechanism of action of FAC in the FAC–MPC composite cement, XRD and SEM-EDS experimental analysis were carried out for MPC paste, FAC paste, and FAC–40MPC paste.

#### 3.5.1. XRD Analysis

The effect of FAC on the hydration products of MPC was examined. The XRD patterns of FAC paste, MPC paste, and MPC–40FAC paste are shown in Figure 6, Figure 7 and Figure 8, respectively.

Figure 6 shows the XRD diffraction spectra of MPC pastes at curing ages of 1 d, 7 days, and 7 + 28 days (water immersion curing for 28 days after standard curing for 7 days) with main diffraction peaks of MgO and KMgPO_4_·6H_2_O (K-struvite). The content of K-struvite at the curing age of 7 days was larger, whereas the corresponding MgO content was lower than that at the curing age of 1 day. The contents of K-struvite and MgO of MPC samples at the curing age of 7 + 28 days decreased, which was in accordance with the findings of Fan [51], because K-struvite and MgO partly dissolved in the MPC samples under the condition of water immersion, verifying that poor water resistance of MPC was due to the part dissolution of K-struvite and MgO after water immersion.

Figure 7 shows the XRD diffraction spectra of FAC pastes at the curing ages of 1 day and 7 days. Crystals, including anhydrous calcium sulfoaluminate (Ca_4_Al_6_(SO_4_)), hydration product of ettringite (Ca_6_Al(SO_4_)_3_(OH)_12_·26H_2_O), calcium carbonate (CaCO_3_), calcium hydroxide (Ca(OH)_2_), dicalcium silicate (Ca_2_SiO_4_), and calcium sulfate (CaSO_4_), were detected in FAC pastes. Wen showed that FAC mineral composition included 4CaO·3Al_2_O_3_·SO_3_, Ca_2_SiO_4_, and a large amount of iron phase (C_6_AF_2_) [38]. Because of the existence of C_6_AF_2_ in FAC, part of Al_2_O_3_ in aluminum hydrate could be substituted by Fe_2_O_3_. Thus, the hydration products mainly included C_3_(A,F)·3CaSO_4_·32H (with the abbreviation of AFt), AH_3_ gel, C–S–H gel, FH_3_ gel, and Ca(OH)_2_, as well as a small amount of C_3_(A,F)·3CaCO_3_·12H (with the abbreviation of AFm) and C_4_(A,F)H_13_. However, the hydration products of AH_3_ gel, C–S–H gel, and FH_3_ gel with the hydration product form of an amorphous gel, which belonged to the amorphous phase, could not be detected by XRD.

Figure 8 shows the XRD diffraction spectra of FAC–40MPC pastes at curing ages of 1 d, 7 days, and 7 + 28 days (water immersion for 28 days after standard curing for 7 days). This showed that the main hydration products of FAC–40MPC pastes included KMgPO_4_·6H_2_O (K-struvite), Ca_6_Al(SO_4_)_3_(OH)_12_·26H_2_O, CaCO_3_, Ca(OH)_2_, Ca_2_SiO_4_, and CaSO_4_. Thus, in FAC–40MPC pastes, the hydration of FAC and MPC influenced each other, whereas they were independent of each other to some extent. The formation of the new hydration products was not confirmed by XRD test results. However, at least, despite the formation of new compounds, the amount was little or in the form of gel [22,27,43].

#### 3.5.2. SEM-EDS Analysis

The SEM test results of MPC pastes and MPC–40FAC pastes are shown in Figure 9 and Figure 10, respectively. The corresponding EDS test results are shown in Figure 11. The microstructure of MPC pastes at the curing age of 1 day is shown in Figure 9a. The MPC hydrate had a porous, layered, interconnected crystal structure. However, the MPC hydrate at the curing age of 7 days (Figure 9b) had a more compact structure compared with that at the curing age of 1 day. Besides, a large amount of rod-like K-struvite was generated in the relatively larger pore interior. The microstructure of MPC–40FAC pastes at the curing ages of 1 day and 7 days (Figure 10a,b) was more compact compared with that of the MPC pastes at the same curing age. A large number of mutual agglutinate particles and flocculent substances were present on the surface of MPC–40FAC at the curing age of 1 d, whereas the surface of MPC-40FAC was smooth and compact with the filling of tiny particles in pore and fracture.

To analyze the composition of particles and flocculent substances on the surface of MPC-40FAC paste as well as the filling particles in fracture, EDS analysis was carried out for Areas 2 and 3 (shown in Figure 11b,c). The results revealed that Area 2 was rich in O, Ca, C, Al, and S with corresponding atom percentages of 58.56%, 22.94%, 10.84%, 4.94%, and 2.73%, respectively, which were all high-content elements of the FAC hydrate. However, Area 3 also contained a certain amount of Fe element besides the high-content elements of Area 2. Thus, particles and flocculent substances were all formed by the FAC hydration. The addition of FAC could fill the interior gap and fraction of the MPC hydrate, form a protective covering layer on the surface, and thus increase the interior compactness of MPC.

The microstructure of MPC at the curing age of 7 + 28 days (water immersion for 28 days after standard curing for 7 days), as shown in Figure 9c, revealed that MgO and K-struvite in the MPC hydrate were massively dissolved after water immersion curing for 28 days. Large-scale crystals in the hydrate dissolved in small-scale crystals with needle-and rod-like alternate arrays because of the corrosion of water. To analyze the residual material composition after water immersion of MPC, the EDS analysis was carried out for Area 1 (Figure 11a). The results revealed that Area 1 was rich in B, O, Mg, P, and K with corresponding atom percentages of 21.38%, 50.92%, 11.06%, 6.21%, and 10.34%, respectively, wherein the ratio of atoms between K and Mg was close to 1:1, which was the theoretical ratio of K-struvite. Thus, the poor MPC water resistance and the decreased strength after water immersion was due to a large amount of gap in the sample interior resulting from the massive dissolution of MgO and K-struvite in the MPC hydrate of water immersion curing, which was in accordance with previous findings [3].

The microstructure of MPC–40FAC pastes at the curing age of 7 + 28 days (water immersion for 28 days after standard curing for 7 days) (Figure 10c) showed the compact microstructure, the smooth surface, and the apparent tiny fracture with water immersion curing for 28 days. To analyze the residual material composition on the surface of MPC–40FAC pastes after water immersion curing, the EDS analysis was carried out for Area 4 (Figure 11d). The results revealed the presence of a large amount of high-content elements of B, C, O, AL, Si, S, and Ca as well as a small amount of Fe, Mg, and K. Thus, the MPC hydration product on the surface of the hydrate still dissolved after the water immersion curing of MPC–40FAC, whereas the compact residual compound was mainly composed of the FAC hydration products. The addition of FAC formed a compact protective covering layer on the surface of the MPC hydrate. Under the condition of water immersion curing, the MPC hydration products in the interior of the hydrate of MPC–40FAC pastes were protected from water corrosion, which finally solved the problem of poor water resistance of MPC.

#### 3.5.3. Reaction Mechanism Analysis

Based on the analysis results of XRD and SEM-EDS, FAC–MPC prepared by the substitution of FAC for MgO with the substitution ratio between 30 and 40 wt % had a long coagulation time, high strength, and good water resistance because of the following factors:(a)Compared with the MPC cement, the MPC–FAC composite cement had a low content of MgO, which slowed down the formation of K-struvite in the early phase of hydration and thus prolonged the coagulation time.(b)The addition of FAC cement formed a certain amount of AFt in the MPC–FAC mortar at the curing age of 1 d, which compensated for the decrease in strength because of the decrease in the K-struvite formation and thus realized the early maintenance of strength. Besides, the hydration rate of FAC was relatively slow. In other words, hydration reaction continued to occur after 1 day. Thus, the strength in the late phase could still be increased dramatically, realizing the significant enhancement of strength in the late phase.(c)The water resistance of MPC cement was poor because of the decrease in the strength of the water immersion environment resulting from the K-struvite migration from MPC to water, massive dissolution of K-struvite in the coagulation material, and a large gap in the sample interior. However, cementitious material of AFt and AFm and flocculent colloidal material of AH_3_ gel, C–S–H gel, and FH_3_ gel formed during the hydration of AFt after water immersion of MPC–FAC were filled in the interior gap of the MPC hydrate. These not only increased the interior compactness of the sample but also formed a compact protective covering layer on the surface simultaneously, thus enhancing the water resistance of MPC.

## 4. Conclusions

In this paper, MPC-FAC composite cement was prepared by adding FAC to MPC, and the effect of FAC content on microstructure, solidification time, strength, and water resistance of MPC-FAC composite cement was studied by controlled variable method. It is concluded that the optimum content of FAC in MPC-FAC composite cement ranges from 30% to 40% where MPC–FAC composite cement has the advantages of long setting time, high strength, and good water resistance. The mechanism of FAC in MPC–FAC composite cement system was discussed, which provided a new and effective way for the performance improvement of MPC cement. The following main conclusions can be drawn:(1)The preparation of the MPC–FAC composite cement by the partial substitution of FAC for MgO could prolong the coagulation time of MPC cement and guarantee the strength of the MPC–FAC mortar not to decrease at the curing age of 1 h. With the increase in substitution ratio of FAC for MgO, the coagulation time of MPC–FAC composite cement prolonged. When the substitution ratio of FAC for MgO was 50%, the coagulation time of FAC–MPC composite cement was prolonged to 25 min, corresponding to 67% prolongation compared with that of the MPC cement.(2)With the substitution ratio of FAC for MgO between 30 and 40 wt %, the strength of the MPC–FAC mortar in the early curing phase (1 h and 1 day) did not decrease, whereas the flexural strength and the compressive strength in the late curing phase (3 days, 7 days, and 28 days) increased dramatically compared with those of the MPC mortar at the same age. The maximum composition strength at the age of 28 days increased up to 50%.(3)The flexural strength and the compressive strength did not decrease but increased dramatically for the MPC–FAC mortars with water immersion curing for 7 days or 28 days after standard curing for 7 days, illustrating the better water resistance of MPC–FAC composite cement compared with that of MPC cement.(4)Compared with FAC cement, the MPC–FAC composite cement prepared by the substitution of FAC for MgO could prolong the coagulation time, increase the strength in the late phase, and solve the poor water resistance problem. The mechanism included the decreased reaction rate during the early hydration of MPC because of the existence of hydraulic cementitious material of FAC. And compactness of the interior of MPC–FAC pastes were also increased because of the filling of a large number of cementitious materials, including AFt and AFm, and flocculent colloidal materials, including AH_3_ gel, C–S–H gel, and FH_3_ gel, in the interior gap of the MPC hydrate.

## Figures and Tables

**Figure 1 materials-12-02561-f001:**
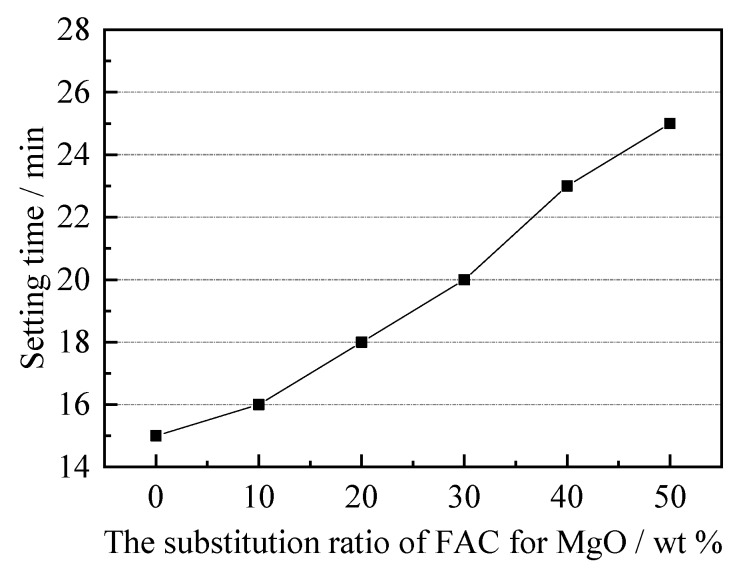
Setting time of fresh magnesium phosphate cement–ferroaluminate cement (MPC–FAC) paste with different FAC content.

**Figure 2 materials-12-02561-f002:**
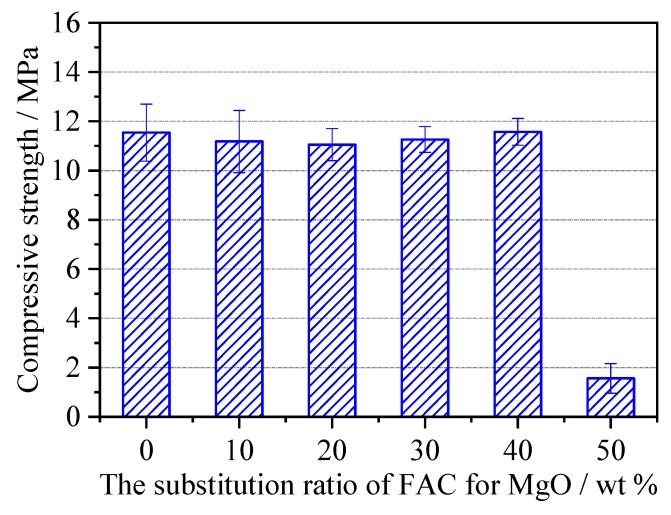
Compressive strength of the MPC–FAC mortar at the age of 1 h.

**Figure 3 materials-12-02561-f003:**
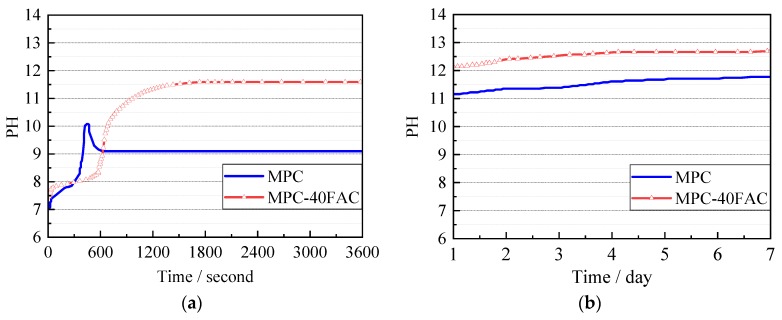
Development of pH values of MPC–40FAC and MPC. (**a**) Early period; (**b**) later period.

**Figure 4 materials-12-02561-f004:**
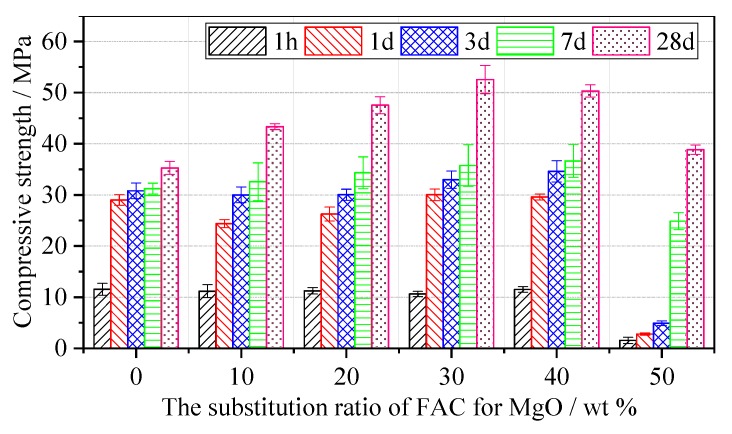
Compressive strength of the MPC–FAC mortar with various dosages of FAC.

**Figure 5 materials-12-02561-f005:**
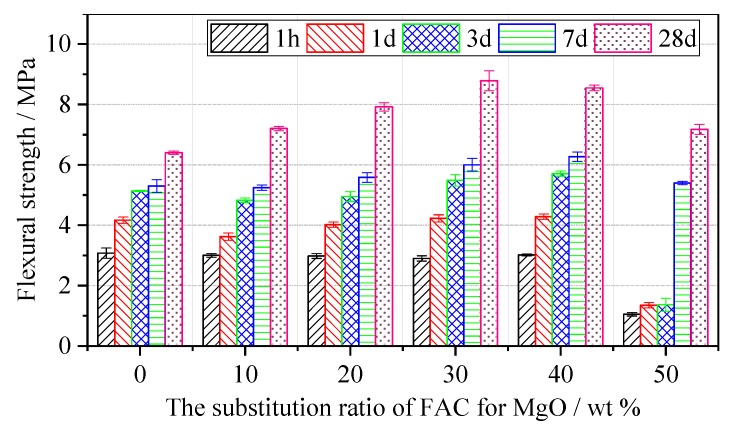
Flexural strength of the MPC–FAC mortar with various dosages of FAC.

**Figure 6 materials-12-02561-f006:**
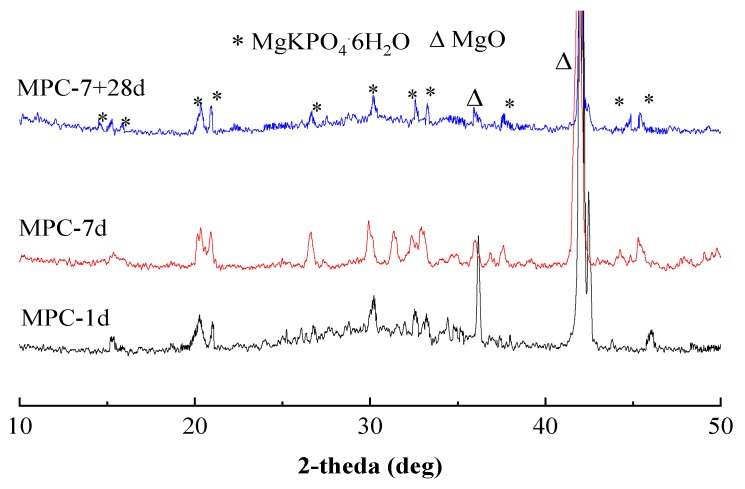
XRD patterns of MPC paste at different curing ages.

**Figure 7 materials-12-02561-f007:**
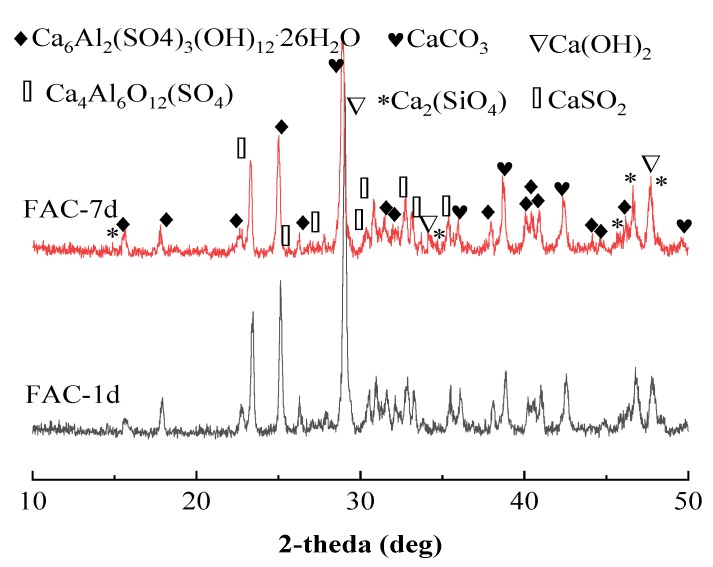
X-ray diffraction (XRD) patterns of FAC paste at different curing ages.

**Figure 8 materials-12-02561-f008:**
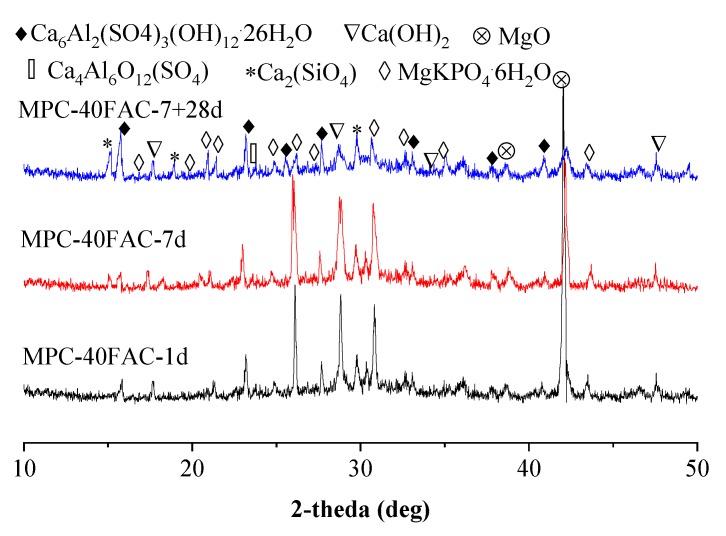
XRD patterns of MPC–40FAC paste at different curing ages.

**Figure 9 materials-12-02561-f009:**
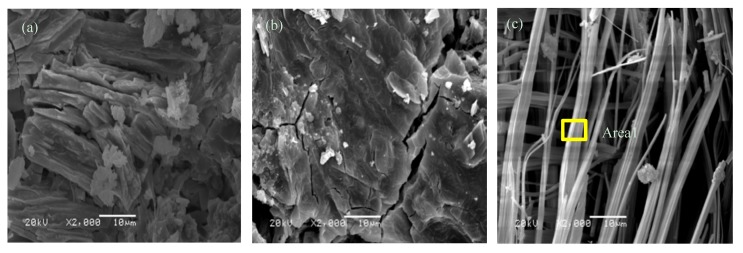
SEM images of MPC composite at different curing ages. (**a**) 1 day; (**b**) 7 days; (**c**) 7 + 27 days.

**Figure 10 materials-12-02561-f010:**
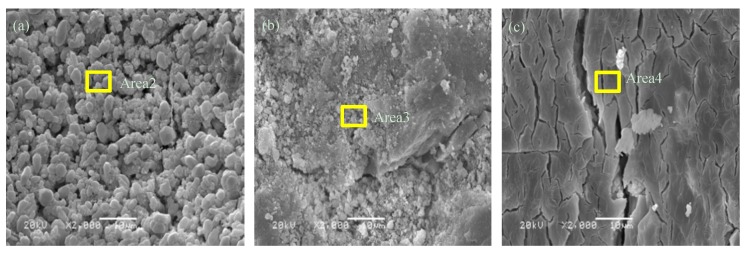
SEM images of MPC–40FAC composite at different curing ages ((**a**) 1 day; (**b**) 7 days; (**c**) 7 + 27 days).

**Figure 11 materials-12-02561-f011:**
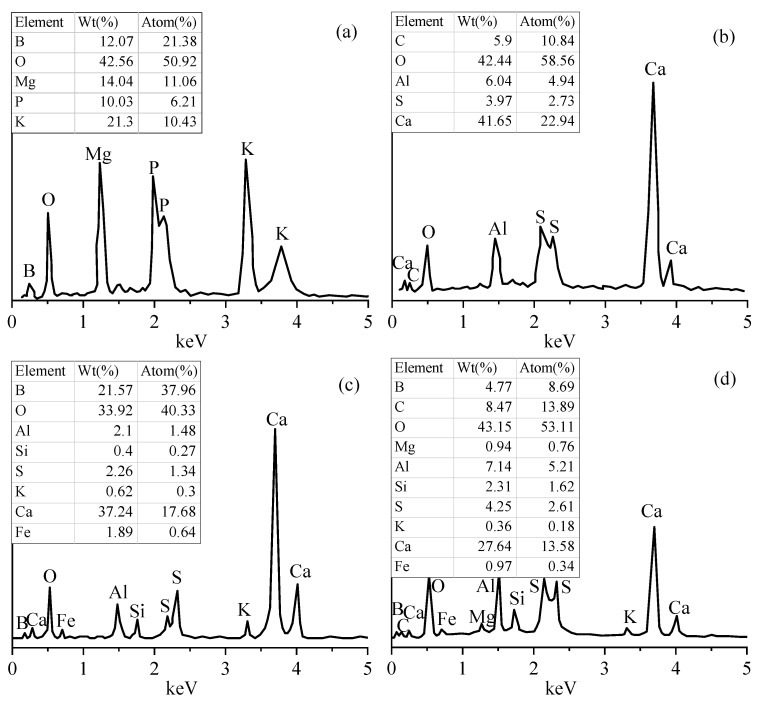
EDS of MPC or MPC–40FAC at different areas. (**a**) Area 1; (**b**) Area 2; (**c**) Area 3; (**d**) Area 4.

**Table 1 materials-12-02561-t001:** Chemical compositions of ferroaluminate cement.

Compound	CaO	SiO_2_	Al_2_O_3_	Fe_2_O_3_	SO_3_	Other
Value (%)	48	6	25	5	13	3

**Table 2 materials-12-02561-t002:** Mix proportions of the sample/wt %.

Mixture ID	FAC/M	PDP/(M + FAC)	BX/B	W/B	S/B
MPC	0	1/3	0.04	0.22	1
MPC–10FAC	1/9	1/3	0.04	0.22	1
MPC–20FAC	2/8	1/3	0.04	0.22	1
MPC–30FAC	3/7	1/3	0.04	0.22	1
MPC–40FAC	4/6	1/3	0.04	0.22	1
MPC–50FAC	5/5	1/3	0.04	0.22	1

Notations: B, M + PDP + FAC, binder; BX, borax; FAC, ferroaluminate cement; M, MgO; PDP, KH_2_PO_4_; S, sand; W, amount of water required for binder mixing and hydration.

**Table 3 materials-12-02561-t003:** Compressive strength retention coefficient of samples.

Mixture ID	7 + 7 Days			7 + 28 Days		
	*F*/MPa	*f*/MPa	*K*	*F*/MPa	*f*/MPa	*K*
MPC	31.23	28.10	0.8998	31.23	24.23	0.7759
MPC–10FAC	32.56	34.58	1.0620	32.56	36.98	1.1357
MPC–20FAC	34.32	39.62	1.1544	34.32	40.34	1.1754
MPC–30FAC	35.74	43.42	1.2149	35.74	45.13	1.2627
MPC–40FAC	36.66	41.58	1.1342	36.66	45.25	1.2343
MPC–50FAC	24.87	27.34	1.0993	24.87	41.18	1.6558

**Table 4 materials-12-02561-t004:** Flexural strength retention coefficient of samples.

Mixture ID	7 + 7 Days			7 + 28 Days		
	*F*/MPa	*f*/MPa	*K*	*F*/MPa	*f*/MPa	*K*
MPC	5.30	4.90	0.9245	5.30	4.22	0.7962
MPC–10FAC	5.24	6.02	1.1489	5.24	6.77	1.2920
MPC–20FAC	5.58	6.18	1.1075	5.58	7.20	1.2903
MPC–30FAC	6.00	6.43	1.0717	6.00	7.59	1.2650
MPC–40FAC	6.27	6.47	1.0319	6.27	7.72	1.2313
MPC–50FAC	5.40	6.48	1.2000	5.40	7.94	1.4704

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
