# Peer review of "Properties and Reaction Mechanisms of Magnesium Phosphate Cement Mixed with Ferroaluminate Cement"

_materials, 2019, doi:10.3390/ma12162561_

Round 1

Reviewer 1 Report

The paper deals with the use ferroaluminate cement (FAC) used as substituent for MgO of during magnesium phosphate cement (MPC) preparation in order to obtain the MPC–FAC composite cement.

The analyzed problem is well described and several experimental results are presented. The presented matter is of secure scientific and technical interest.

I recommend to accept the paper in the present form.

Author Response

Response to Reviewer 1 Comments

Point 1: The paper deals with the use ferroaluminate cement (FAC) used as substituent for MgO of during magnesium phosphate cement (MPC) preparation in order to obtain the MPC–FAC composite cement. The analyzed problem is well described and several experimental results are presented. The presented matter is of secure scientific and technical interest. I recommend to accept the paper in the present form.

Response 1: The authors appreciate the reviewer’s comment.

Reviewer 2 Report

This manuscript describes the properties and reaction mechanisms of magnesium phosphate cement mixed with ferroaluminate cement. Indeed, the addition of FAC to MPC seems to improve the flexural and compressive strengths of MPC-FAC compared to MPC alone.

Here are a few comments regarding the manuscript:

Page 1 - Line 13: I would remove the word "while", or reconstruct the sentence;

P1-L24: Instead of "realize the enhancement of" replace with "enhance the water...";

P2-L37: "Great deals of studies have been attempted";

Table 1: Is it percentage in weight?;

Part 2.2.: For the specimen preparation, was there a specific standard used? How long was the mix stirred for the mortar samples? You used the European standard for the mold size, did you use the European standard for the mixing procedure?;

Part 2.3.: Did you measure both the initial and the final setting times?;

I would replace the word "flexure strength" with "flexural strength" in the manuscript;

Part 2.3.2.: Was the pH measured while the samples were being stirred for 7 days? Or was the pH measured, but the sample was not stirred during the measurements?;

P4-L116: would replace the word "cuboids" with "cubes";

P5-L134: I do not understand the -12 to 164 degrees, do you mean, from 10 degree 2-Theta, to 50 degrees-2 Theta?;

Part 2.3.6.: You mention that samples were dipped in absolute ethanol to stop the hydration, but for how long?;

P9-L217: You mention "the main reason was the more compact structure", but it is not explained here, as you describe them later. It could be a theory at this point, then confirmed by your SEM studies. Have you performed bulk density measurements?;

Figure 6: The triangle is upside down on the graph, which one is right?;

Figure 8: Can you explain why there is a peak of C2S after 7+28 days for MPC-40FAC, while it is not really present before? This phase should actually hydrate to C-S-H, and not form in water?;

P11-L244: What are "coal crystals"?;

P11-L253: You mention that some phases were not detected by XRD, but you could have used TGA to observe the presence of AH3 gel, FH3 gel, ettringite, AFm,...;

Figures 9 and 10: Why did you not show SEM images of FAC samples?

Author Response

Response to Reviewer 1 Comments

Point 1: Page 1 - Line 13: I would remove the word "while", or reconstruct the sentence.

Response 1: The authors appreciate the reviewer’s comment. According to the reviewer’s suggestion, "while" has been removed.

Point 2: P1-L24: Instead of "realize the enhancement of" replace with "enhance the water..."

Response 2: According to the reviewer’s suggestion, "realize the enhancement of" has been replaced with "enhance the water...".

Point 3: P2-L37: "Great deals of studies have been attempted"

Response 3: According to the reviewer’s suggestion, "Great deals of studies have been attempted" has been revised to be "In order to improve the properties of MPC, the researchers conducted a series of studies." in line 38 of Introduction.

Point 4: Table 1: Is it percentage in weight?

Response 4: The authors appreciate the reviewer’s comment on this. Yes, the percentage is weight in table 1. Supplementary explanation could be found in line 89 of revised draft.

Point 5: Part 2.2.: For the specimen preparation, was there a specific standard used? How long was the mix stirred for the mortar samples? You used the European standard for the mold size, did you use the European standard for the mixing procedure?

Response 5: Thanks for the comments. The specimen preparation and the mixing procedure uses Chinese standards (GB/T 17671-1999[41]), using a cement mortar mixer to mix the dry mixture for 30s, adding water and stirring for 90s.

Point 6: Part 2.3.: Did you measure both the initial and the final setting times?

Response 6: The authors thank the reviewer for the valuable assessment; since the interval time of MPC between the initial setting time and the final setting time is very short, the initial setting time was taken as the setting time. The authors have revised, the detailed revision could be found in Page 4 of part 2.3.1. 

Point 7: I would replace the word "flexure strength" with "flexural strength" in the manuscript.

Response 7: According to the reviewer’s suggestion, the word "flexure strength" has been replaced with "flexural strength" in the manuscript.

Point 8: Part 2.3.2.: Was the pH measured while the samples were being stirred for 7 days? Or was the pH measured, but the sample was not stirred during the measurements?

Response 8: Thanks for the comments. The binder was mixed with water, and after stirring uniformly, the change in pH was recorded with a pH meter. After the cement paste hardened, the supernatant of the sample is taken for measurement (the process is not stirred).

Point 9: P4-L116: would replace the word "cuboids" with "cubes".

Response 9: Thanks for the comments. The authors think the word "cuboids" would not replace with "cubes". Because the MPC–FAC mortar samples’ size is 40 × 40 × 160 mm3.

Point 10: P5-L134: I do not understand the -12 to 164 degrees, do you mean, from 10 degree 2-Theta, to 50 degrees-2 Theta?

Response 10: Thanks for the comments. The -12 to 164 degrees is a device parameter of the XRD-7000X diffractometer (Shimadzu Co., Japan), which means that its maximum radiation in the 2θ range is from -12 to 164. The scan range selected in this experiment is 2θ =10°-70°, and other range diffraction peaks are weak.

Point 11: Part 2.3.6.: You mention that samples were dipped in absolute ethanol to stop the hydration, but for how long?

Response 11: The authors appreciate the reviewer’s comment; the samples were immersed in absolute ethanol for 24 h to stop the hydration, the detailed revision could be found in line 147 of 2.3.6.

Point 12: P9-L217: You mention "the main reason was the more compact structure", but it is not explained here, as you describe them later. It could be a theory at this point, then confirmed by your SEM studies. Have you performed bulk density measurements?

Response 12: Thanks for the comments. It is a pity that I have not done volume density measurement, but I will make further improvement in the follow-up research. The analysis of "the most important reason is the more compact structure" is based on the scanning electron microscope results, which is consistent with the results of zhang et al.'s study. (Reference cited: Ge Zhang, Guoxin Li, Tingshu He. "Effects of sulphoaluminate cement on the strength and water stability of magnesium potassium phosphate cement", Construction and Building Materials, 2017)

Point 13: Figure 6: The triangle is upside down on the graph, which one is right?

Response 13: According to the reviewer’s suggestion, the authors have modified the triangle on the graph and the upside triangle is correct. The detailed revision could be found in Figure 6.

Point 14: Figure 8: Can you explain why there is a peak of C2S after 7+28 days for MPC-40FAC, while it is not really present before? This phase should actually hydrate to C-S-H, and not form in water?

Response 14: The authors appreciate the reviewer’s comment. The peak of C2S in the MPC-40FAC sample does not appear after 7 + 28D, but is generated in the early stage of MPC-40FAC hydration (the early stage C2S peak is weak), as shows in Figure 8. (Note: the peak position in the figure does not only represent 7 + 28D product, each label also represents the product of the corresponding peak position in the 1d and 7d samples.) This phase is indeed water-synthesized C-S-H, and its reaction equation is

2CaO  ·SiO2+2H2O → Ca-SiO2-H2O + Ca(OH)2.

Point 15: P11-L244: What are "coal crystals"?

Response 15: Thanks for the comments. It was just a slip of the pen, and "coal crystals" should be "crystals". The detailed revision could be found in line 257 of revised draft.

Point 16: P11-L253: You mention that some phases were not detected by XRD, but you could have used TGA to observe the presence of AH3 gel, FH3 gel, ettringite, AFm,...

Response 16: Thank you very much for your advice. You provided me with a very good test method, but due to the limited test conditions, it is difficult to make up this part in a short time. Therefore, according to the current test and conditions, this paper can only carry out inferential analysis of product composition according to the existing research results [38, 39, 40]. The TGA test will be carried out in further research to further confirm the product composition of the composite cement.

Point 17: Figures 9 and 10: Why did you not show SEM images of FAC samples?

Response 17: Thank you very much for your advice. You provided me with a very good test method, but due to the limited test conditions, it is difficult to make up this part in a short time. Therefore, according to the current test and conditions, this paper can only carry out inferential analysis of product composition according to the existing research results [38, 39, 40]. The TGA test will be carried out in further research to further confirm the product composition of the composite cement.

Reviewer 3 Report

The paper deals with the selected properties of FAC and substitution by MgO up to 40 wt.%. Used procedures are clearly described and experimental design is correct. The only one weakness of the paper, which considerably reduces the overall rating, is the quality of discussion, respectively, there are described obtained results in the paper, however they are discussed in relation to those publisek works. That should be improved to amplify the novelty and the contribution to current knowledge. After major revision I recommend to accept.

Author Response

Point 1: The paper deals with the selected properties of FAC and substitution by MgO up to 40 wt.%. Used procedures are clearly described and experimental design is correct. The only one weakness of the paper, which considerably reduces the overall rating, is the quality of discussion, respectively, there are described obtained results in the paper, however they are discussed in relation to those publisek works. That should be improved to amplify the novelty and the contribution to current knowledge. After major revision I recommend to accept.

Response 1: Thanks for the comments. In this paper, based on XRD, SEM and EDS test results, the improvement mechanisms of mechanical properties and water resistance of MPC-FAC composite cement system were theoretically analyzed from two aspects of product analysis and microstructure morphology. The mechanism of action in the MPC-FAC composite cement system was compared with published works to verify the correctness of the experimental results and analytical theory (line304-307) (line260-261).

The innovation of the paper is that the more iron phase content in the FAC (line72-75) makes the MPC-FAC system have a high pH value, which contributes to form AFt and a series of colloidal substances in the system. Therefore, the internal compactness of the sample increased, while the compact protective covering layer was generated on the surface to realize the enhancement of water resistance and strength in the late curing time (abstract). This method expands the application range of MPC cement and provides a novel and effective way to improve the mechanical properties and water resistance of MPC cement.

According to the recommendations of the reviewers, in order to improve the novelty and the contribution to current knowledge, the authors have carefully modified the conclusions. The detailed revision could be found in the line 347-354 of Conclusions of revised draft: “In this paper, MPC-FAC composite cement was prepared by adding FAC to MPC, and the effect of FAC content on microstructure, solidification time, strength and water resistance of MPC-FAC composite cement was studied by controlled variable method. It concluded that the optimum content of FAC in MPC-FAC composite cement ranges from 30% to 40% where MPC-FAC composite cement has the advantages of long setting time, high strength and good water resistance. The mechanism of FAC in MPC-FAC composite cement system was discussed, which provided a new and effective way for improving the performance of MPC cement. The following main conclusions can be drawn:”.

Round 2

Reviewer 3 Report

The paper is very novel and uncertainties were explained. Hence, I recommend acceptation of the paper in its current form.